# Omics and Artificial Intelligence to Improve In Vitro Fertilization (IVF) Success: A Proposed Protocol

**DOI:** 10.3390/diagnostics11050743

**Published:** 2021-04-21

**Authors:** Charalampos Siristatidis, Sofoklis Stavros, Andrew Drakeley, Stefano Bettocchi, Abraham Pouliakis, Peter Drakakis, Michail Papapanou, Nikolaos Vlahos

**Affiliations:** 1Second Department of Obstetrics and Gynecology, “Aretaieion Hospital”, Medical School, National and Kapodistrian University of Athens, Vas. Sofias 76, 11528 Athens, Greece; mixalhspap13@gmail.com (M.P.); nikosvlahos@med.uoa.gr (N.V.); 2Assisted Reproduction Unit, Second Department of Obstetrics and Gynecology, “Aretaieion Hospital”, Medical School, National and Kapodistrian University of Athens, Vas. Sofias 76, 11528 Athens, Greece; 3Assisted Reproduction Unit, First Department of Obstetrics and Gynecology, Medical School, National and Kapodistrian University of Athens, Alexandra Hospital, 80 Vas. Sofias Av. and Lourou str., 11528 Athens, Greece; sfstavrou@yahoo.com (S.S.); pdrakakis@med.uoa.gr (P.D.); 4Hewitt Fertility Centre, Liverpool Women’s NHS Foundation Trust, Crown Street, Liverpool L8 7SS, UK; adrakeley@yahoo.com; 5Second Unit of Obstetrics and Gynecology, Department of Biomedical and Human Oncologic Science, Policlinico University of Bari, 70124 Bari, Italy; stefanoendo@tin.it; 6Second Department of Pathology, National and Kapodistrian University of Athens, “Attikon” University Hospital, Rimini 1, Chaidari, 12642 Athens, Greece; apouliak@med.uoa.gr

**Keywords:** in vitro fertilization, assisted reproductive techniques, metabolomics, transcriptomics, microRNAs, artificial intelligence, artificial neural network

## Abstract

The prediction of in vitro fertilization (IVF) outcome is an imperative achievement in assisted reproduction, substantially aiding infertile couples, health systems and communities. To date, the assessment of infertile couples depends on medical/reproductive history, biochemical indications and investigations of the reproductive tract, along with data obtained from previous IVF cycles, if any. Our project aims to develop a novel tool, integrating omics and artificial intelligence, to propose optimal treatment options and enhance treatment success rates. For this purpose, we will proceed with the following: (1) recording subfertile couples’ lifestyle and demographic parameters and previous IVF cycle characteristics; (2) measurement and evaluation of metabolomics, transcriptomics and biomarkers, and deep machine learning assessment of the oocyte, sperm and embryo; (3) creation of artificial neural network models to increase objectivity and accuracy in comparison to traditional techniques for the improvement of the success rates of IVF cycles following an IVF failure. Therefore, “omics” data are a valuable parameter for embryo selection optimization and promoting personalized IVF treatment. “Omics” combined with predictive models will substantially promote health management individualization; contribute to the successful treatment of infertile couples, particularly those with unexplained infertility or repeated implantation failures; and reduce multiple gestation rates.

## 1. Introduction

Infertility is currently characterized as a disease with variable socioeconomic extends. To date, in vitro fertilization (IVF) success rates remain relatively low, ranging from 4 to 40% based on various parameters, mainly patients’ age [1,2]. As meticulously reported in a six-year-old paper, “infertility remains a highly prevalent global condition and is estimated to affect between 8 and 12% of reproductive-aged couples worldwide, with 9% currently cited as the probable global average” [1]. Moreover, in some regions of the world, especially in Europe, Africa, and Asia, the rates of infertility are much higher, reaching ~30% in some populations [1,2]. Of note, the real number worldwide is difficult to assess, as there is heterogeneity in both the criteria used to define the disease and in the types of studies employed. We also estimate that this is due to an inaccurate cumulative assessment of couples’ reproductive characteristics; the IVF cycle itself; and the lack of incorporating indications from certain biological aspects, especially molecular investigations and “omics”.

The prediction of IVF outcome would be an imperative achievement in assisted reproduction, substantially aiding infertile couples; health systems and communities; health policies; and insurance and guideline construction authorities, which are efficient in terms of cost, time and minimizing the emotional stress of infertile couples linked with these procedures.

There is a general tendency towards a fully automated performance of clinical analytical protocols and/or management to downstream the overall errors and upstream the quality/quantity of performance under optimal conditions [3]. Here, the modern IVF Unit is not an exception; therefore, several technologies have been proposed, and even though they have not yet been fully assessed to be established as reliable approaches in the clinical management of infertility, the interest is growing.

As an example, the metabolomic profile of the embryo and transcriptomics of the oocyte and endometrium provide instant and accurate information on the embryo’s ability for implantation and progression to live birth [4,5,6,7]. Quoting the paper of Altmäe et al. (2014), “Omics” refers to the application of high-throughput techniques that simultaneously take into consideration the alterations in the genome, epigenome, transcriptome, proteome or metabolome in a certain biological sample [6]. There are also new fields in biological data, such as exomics, lipidomics and secretomics. In this context, some of the numerous examples in the literature are reports centering on creating molecular tools using microarray technology (ERA), on determining the transcriptomic signature of the endometrium during the window of implantation: authors found that 18 of the 25 gene targets were included in the ERA design [8]. Therefore, “omics” data are a valuable parameter for embryo selection optimization and promoting personalized IVF treatment [8]. “Omics” combined with predictive models will substantially promote health management individualization; contribute to the successful treatment of infertile couples; particularly those with unexplained infertility or repeated implantation failures; and reduce multiple gestation rates.

We propose a combination of statistical models with novel and flexible artificial neural network (ANN) architectures and conformed input and output parameters according to the clinical and bibliographical standards, driven by a complete data set and “trained” by a network expert. The aim is to enhance the probability of offering optimal management and consultation to infertile couples who have failed to conceive after an IVF cycle. Moreover, the combination of data with this radical technological development will allows us to integrate complex systems in healthcare and employ automated systems as a supporting structure in decision making. These mathematical networks when appropriately built and adequately trained may allow a more objective course by implementing all the significant parameters. They are capable of utilizing and evaluating a vast amount of information in a rapid and automated manner to provide an objective indication on the outcome of an IVF cycle. 

Our proposal aims to incorporate the application of artificial intelligence in the evaluation of individual and combined fertility status, as well as to predict the outcome of an upcoming IVF attempt, through a combination of data, analyzing them using modern mathematical techniques. In this respect, we will combine a variety of fertility-related information to stratify couples that are at risk into groups and apply the optimal treatment to increase the probability for a successful outcome; moreover, our project goes one step further, as detailed information for each couple will be used in a personalized manner.

## 2. Objectives

Our proposal aims to develop a novel tool, integrating omics, demographic and lifestyle information to:
Evaluate individual and combined fertility status, so as to predict the outcome of an upcoming IVF attempt after a failed cycle;Propose optimal treatment options and enhance treatment success rates.

For this purpose, we will proceed with the following:
Recording infertile couples’ lifestyle and demographic parameters;Recording infertile couples’ previous IVF cycle characteristics;Measurement and evaluation of metabolomics, transcriptomics and biomarkers;Development of mathematical models to assess the usefulness of the collected data in generating predictive values in the subsequent cycles and the stratification of the involved population at risk;Creation of ANN models to increase objectivity and accuracy in comparison to traditional techniques for the improvement of the success rates of IVF cycles following an IVF failure.

## 3. Materials and Methods

### 3.1. Lifestyle, Demographic and Previous Cycle Characteristics

The group of lifestyle and demographic characteristics will comprise the following parameters: age (male/female), BMI (male/female), smoking/alcohol (male/female), duration of infertility, primary or secondary infertility and type of infertility. In the group of previous cycle characteristics, the following parameters will be included: anti-Mullerian hormone (AMH); antral follicle count (AFC); follicle stimulating hormone (FSH); thyroid stimulating hormone (TSH); sperm parameters, including DNA fragmentation, oxidation reduction potential and artificial intelligence (AI); assessment of the sperm used for Intracytoplasmic Sperm Injection (ICSI); number of previous IVF attempts, gonadotrophin dosage; number of high-quality embryos/blastocysts; hysteroscopic findings. The suggested parameters are based on a previous work of the same team [3].

### 3.2. Metabolomic Profile

The metabolomic profile of each embryo represents its functional phenotype; is entirely non-invasive, thus safe, regarding the embryo; and represents cell physiology and subsequently its implantation potential [4,9,10,11]. The rationale for the value of the embryos metabolomic profiling starts from the current lack of understanding to determine the embryos with the highest reproductive potential, which is likely to contribute to the high rate of implantation and IVF cycle failures. Several reports have demonstrated that analyzing all small molecules used and metabolized by the embryo [12,13,14,15,16,17] can help in the most appropriate embryo selection. On the other hand, robust answers from high-quality research cannot be pooled to date [18]. 

The parameters in the culture media from embryos to be measured include specific or single biomarkers (pyruvate, glucose, amino-acids, oxygen and leptin) and endogenous metabolites of different classes (acyl carnitines; amino acids; hexose; sphingolipids; glycerophospholipids; biogenic amines; steroid hormones: mineralocorticoids, glucocorticoids and sex steroids) [19,20,21]. The goal is to identify a metabolic signature that, combined with other “omics” data, allows the selection of couples who will undergo IVF with an increased probability of a successful outcome and allow personalized healthcare.

### 3.3. Transcriptomics of the Oocyte

The concept behind transcriptomics is that gene expression in cumulus cells (CCs) engulfing the oocyte provides crucial information on the competence of the oocyte itself and, furthermore, the embryo, as these two differential cell types grow and develop in a highly coordinated and mutually dependent manner. Gene expression analysis of CCs is achieved through the isolation of these cells before ICSI or IVF and the analysis of the extracted mRNA through microarray analysis, RT-PCR or quantitative RT-PCR to obtain its transcriptomic profile [22,23]. A recent study, among many others, showed the connection between the CCs transcriptomic profile and embryo quality. Authors reported that the differential gene expression profile of human CCs, including MSTN, CTGF, NDUFA1, VCAN, SCD5 and STAR, can be used as potential indicators of embryo quality [24]. Thus, the parameters analyzed provide an indication of the competence and quality of the oocyte through positive and negative association with the gene expression profile and, furthermore, may act as positive/negative predictors of the embryo quality itself and pregnancy outcome. 

In this context, we propose some useful examples from data deriving from the current literature: selected biomarkers (PCK1, BCL2L11, NFIB, CCND2, CXCR4, GPX3, HSPB1, DVL3, DHCR7, CTNND1, TRIM28, HAS2, GREM1, STAR, AREG, Cx43, PTGS2, SCD1, SCD5, HAS2, PTGS2 and BDNF) are categorized according to the detection method and measurement outcome, and the final set examined will be defined by the method of choice and, moreover, by assessing its importance in indicating oocyte competence and embryo quality.

### 3.4. Transcriptomics of the Endometrium

The endometrium represents a dynamic tissue component characterized by cyclic hormonal changes reflecting cyclic changes in gene expression during the menstrual cycle. Endometrial transcriptomics are an emerging “omics” potential for the clarification of the mechanisms lying behind endometrial pathophysiology and recurrent implantation failure. The genes actively expressed in a specific population of cells is reflected via the transcriptome. A battery of studies has unraveled many simultaneously up- and down-regulated genes involved in the acquisition of embryo implantation and endometrial receptivity [25]. Transcriptomic analysis with microarrays or RNA-seq using biopsy of endometrium tissue targets the identification of the genes that regulate endometrial receptivity during the “implantation window” of the cycle. A number of genes over expressed during the “implantation window” have recently been identified (SPP1, IL15, MFAP5, ANGPTL1, EG-VEGF NLF2, LAMB3) [26,27]. The incorporation of transcriptomics in predictive molecular tests for endometrial receptivity appears to be a novel, powerful tool that could make the personalized embryo transfer feasible [8].

### 3.5. Reactive Oxygen Species (ROS) in Follicular Fluid

Reactive Oxygen Species (ROS) are produced as a normal product of aerobic metabolites. In an IVF setting, the existing literature suggests a favorable outcome in terms of oocyte quality/maturation and fertilization rate with increased ROS levels, while other studies report significant data on the detrimental effect of increased ROS concentration in the quality of embryos exposed and their potential to advance [28,29,30,31]. A recent study in women with polycystic ovary syndrome showed high oxidative stress in both follicular fluid and serum concentration markers: such markers in the former seemed to be directly associated with and thus predict embryo quality in IVF [32]. For this assessment, chemiluminescence could be assigned as the preferred detection method, whereas we initially intend to employ ROS cut-off levels of 100-cps/400 μL of follicular fluid. We anticipate that these data will clarify their role by defining if ROS products are beneficial or not.

### 3.6. MicroRNA (miRNA) of the Endometrium

MicroRNAs (miRNAs) are detected in the underlying molecular activities that surround implantation and further progression of the embryo. Since endometrial receptivity is a key component in the process of reproduction and miRNAs, to this end provide vital information in the regulation of gene expression responsible for the implantation process, we aim to signify the presence of miRNAs in endometrial tissue samples, obtained through hysteroscopy and sampling of the endometrium (pipelle/endogyn) during the implantation window (day 21 to 24) or early follicular phase (day 6 to 9), in the cycle preceding that of the IVF treatment. For the analysis, total RNA is proposed to be extracted from the biopsy samples and amplified/quantified through real time PCR, while miRNA signature will be determined through microarrays. The proposed miRNAs to be studied are as follows: hsa-miR-30b, -30d, -26b, -21, -494, -10a, -923_v12.0 and hsa-let-7g [33].

### 3.7. Artificial Neural Networks (ANNs) and Other Mathematical Models

Using a novel technological development through the integration of AI, through ANNs and conventional mathematic techniques, complementing each field, and fed and tested from the combination of data that “omics” and individual characteristics of infertile couples, we aim to develop a predictive and decision-supporting tool. 

Various types of models are eligible: we know that ANNs exert a better performance than classical statistical analysis [34,35], but they have limitations in determining important prediction factors and the degree of their impact on the final results [36]. We aim to use two families of techniques: Supervised and unsupervised ANNs, such as the learning vector quantizer, the back propagation and the self-organizing map, for the estimation of the repeated IVF cycle outcomes, and especially for three cut-off points/outcome parameters (clinical pregnancy, miscarriage and live birth);Parametric and non-parametric statistical techniques, such as regression analysis and classification and regression trees, in order to determine the influence of each variable to the final outcome.

We have two final goals: first, the prediction of the IVF outcome at various steps and for the above-mentioned end points, and second, a more accurate consultation with the couple after their first IVF cycle for the next steps. Of note, an IVF result prediction accuracy at the level of 75% is currently considered successful, according to the available bibliographic data [35]. The construction of the proposed models will be based on two types of data:Couples’ data that are retrospectively collected and provided by all IVF units;New data collected (see categories above, mostly related to omics).

Data will be used both retrospectively and prospectively, in the sense that all new data will serve to continually feed the central database and to synchronously improve the system accuracy. 

Furthermore, images of the oocytes, sperm and embryos will be collected and evaluated using AI deep machine learning, as recent studies have suggested their assessment could better inform embryo selection [37].

### 3.8. Multitier Distributed and Flexible Information Technology System for Various IVF Units

According to the information and communication technology component of this proposal, we could develop a multitier distributed and flexible information technology system to support an arbitrary number of IVF units. This is envisioned to have a centralized logic and intelligence, along with the capability to interact with remote databases to retrieve and exploit anonymized data from partner databases [38].

### 3.9. Further Innovative Steps across IVF Units

A further aim involves the construction, evaluation, production and utilization of various IVF outcome prediction models, as well as the extraction of knowledge from the available data for individual omics biomarkers and for their combination. The end (system) users will be the fertility treatment providers in the IVF units. For each unit, a separate interface supported by the central infrastructure will be available, while each IVF unit will act independently, securely isolated from the others. At the same time, IVF units will be able to use data from other sources feeding the central database, while these data will serve as feedbacks for the creation of more powerful models, in terms of accuracy.

### 3.10. Team Cooperation—Main Goals

We aim to develop an initiative for establishing the co-operation of such multidisciplinary fields for the development of a novel intelligent “omics”-based system in the assessment of infertile couples. This project is designed to integrate the experience of established IVF laboratories, the academic prestige and background knowledge from university partners, the cutting-edge technology and experience in “omic” investigation from the respective institutes, aligned with the expertise of specialists in data handling, analysis and software development. For the realization of this project, experienced laboratories throughout Europe will apply cutting-edge technologies in “omics” investigation, and more specifically, in the metabolomics and transcriptomics area, in order to acquire valuable information on the expression of specific cellular products and establish their relations with fertility and reproduction. The validity of these data will be ensured by performing a multilevel analysis to establish their relationship with fertility status, by combining other medical and lifestyle factors affecting fertility and testing their significance in predicting IVF outcome and eventually through the evaluation of the produced system and its efficiency by the respective IVF laboratories, assessing its utility. The final stage of this project will be its cost-beneficial and cost-effective implementation by a health economist.

This project is designed to conform with all the ethical, legal and regulatory aspects of Assisted Human Reproduction, as designated by the National and European authorities. The sum of the participating laboratories and clinics operate strictly in accordance with the National and European directives and regulations, and the respective National Ethics Committees ensure that the protocols followed comply with the ethical aspects surrounding gamete/embryo handling and the handling of biological material for research purposes. Since legal and ethical regulation is an absolute requisite for an IVF or research unit/laboratory to operate in the European Region, combined with the stated prospects of this project that do not aberrate from the procedures already in practice, there are no implications from an ethical, legal and regulatory viewpoint. In terms of cost effectiveness, an experienced partner in health economics undertakes this important task, for the evaluation of the proposed methodology along with the anticipated increased IVF success rate, as well as avoidance of multiple gestation (among other implications) for each country separately, taking into account the diversity in previously applied methods as well as costs of treatment and applied funding policies by health insurance organizations.

A concept diagram of the proposed protocol is presented in Figure 1.

## 4. Conclusions

To date, various predictive models using demographic and previous cycle parameters, and the evaluation of either metabolomics or transcriptomics, have been evaluated for optimizing IVF outcome, but separately, thus preventing the necessary desired robust results. We propose a system aiming to overcome the lack of accuracy such systems are currently presented with. This is due to the fact that we will combine personal and lifestyle infertile couples’ data from different European populations with the most commonly used and accessed “omics” (transcriptomics and metabolomics), interpreted and evaluated through modern mathematical techniques, especially ANNs. After the application of statistical models, the system will be capable of distinguishing, among a large number of demographics, clinical and biochemical parameters, those with the greatest significance for predicting the outcome of an upcoming IVF cycle. Thus, a single device integrating both the molecular tests as well as the outcome of the intelligent systems from the previous cycle could be built. The proposed system is expected to combine high accuracy and efficacy, thus leading to cost effectiveness and benefits to the society and economy. Eventually, we expect the evolution of a simple, reproductive and cheap system that could be used in everyday practice in an IVF unit, especially after the large-scale evaluation of metabolomics and transcriptomics, which will decrease test costs, thus making them available for routine use. These improvements are to be universally adopted, as this system is expected to receive a wide acceptance as an innovative managerial tool in fertility centers worldwide and could possibly be established in routine clinical practice for the evaluation and management of infertile patients in the near future.

Especially, data will be used both retrospectively and prospectively, in the sense that all new data will serve to continually feed the central database and to synchronously improve the system accuracy. The “Omics” concept will be divided into three phases: 1. analysis of a first sample cohort with different techniques; 2. integration of data and identification of a marker panel for validation; and 3. validation of markers identified during discovery. 

We have two final goals: first, the prediction of the IVF outcome at various steps using omics, and second, improved accuracy at the follow-up consultation with the couple after their first IVF failure, providing realistic expectations and informing their next steps to take. The anticipated impact can be divided into two axes: 1. to demonstrate omics’ potential to guide infertility management and 2. to constitute an inexpensive, widely used IVF tool representing an escalated and stratified approach, replacing conventional methods.

## Figures and Tables

**Figure 1 diagnostics-11-00743-f001:**
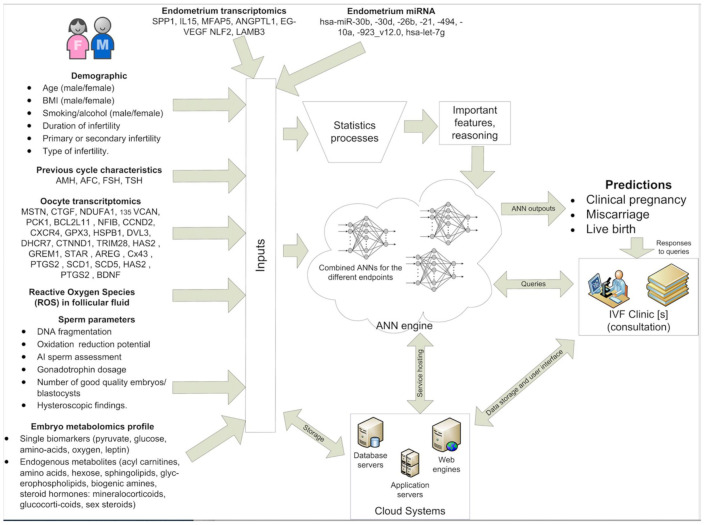
Concept diagram of the proposed protocol. Inputs (**left side**) of the proposed demographics, previous IVF data, and omics characteristics are used by: (a) the statistics processors to extract important features and lead to a reasoning and (b) by the various ANNs to predict (i) clinical pregnancy, (ii) miscarriage and (iii) live birth. The ANN system should be able to be used by numerous IVF clinics. IVF clinics issue queries for specific couples and obtain as response the ANN outputs at the three prediction levels. Finally, the system is supported by a multitier cloud system composed of database, application and web servers, which are responsible for: (a) storing patient data; (b) running the ANNs and the user logic; (c) presenting the results through a web interface.

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
