# Peer review of "Omics and Artificial Intelligence to Improve In Vitro Fertilization (IVF) Success: A Proposed Protocol"

_diagnostics, 2021, doi:10.3390/diagnostics11050743_

Round 1

Reviewer 1 Report

In this paper, the –omics approach combined with the benefits of artificial intelligence is used to perform the enhanced in-vitro fertilization.

The work dealing with IVF focuses on the aid to the infertile couples and the righteous treatment of this issue.

INTRODUCTION (note)

Please, provide some statistical data (for example European, worldwide numbers of couples suffering from this issue) as for infertility problem and describe in brief this problem.

INTRODUCTION (note 2)

Please, in brief describe character and function, and current status of the “-omics” approach.

INTRODUCTION (upgrade) – This papers I consider important to cite

There is a general tendency towards a fully automated performance of clinical analytical protocols and/or management to downstream the overall errors and upstream the quality/quantity of performance under optimal conditions [https://doi.org/10.1016/j.cca.2020.04.015]. Here, the modern IVF Unit is not an exception, therefore several technologies have been proposed and even though they have not yet been fully assessed to be established as reliable approaches in the clinical management of infer-tility, the interest is growing [3].

MATERIALS AND METHODS (note)

Please, draw a workflow scheme on the proposed protocol based on coupling the omics with artificial intelligence.

Author Response

The work dealing with IVF focuses on the aid to the infertile couples and the righteous treatment of this issue.

We would like to thank the reviewer for taking the time and effort to assess our original submission so meticulously. We have taken into account all of your comments and recommendations and we have modified our paper accordingly. All manuscript changes have been highlighted using the “tracked changes” function provided by Microsoft Word. Detailed replies to the reviewer’s comments are provided below:

INTRODUCTION (note)

Please, provide some statistical data (for example European, worldwide numbers of couples suffering from this issue) as for infertility problem and describe in brief this problem.

A: We thank you for the comment. We have added some more sentences in the introduction section; it now reads: “As meticulously reported in a six-year-old paper, “infertility remains a highly prevalent global condition and is estimated to affect between 8 and 12% of reproductive-aged couples worldwide, with 9% currently cited as the probable global average” [1]. Moreover, in some regions of the world, especially in Europe, Africa, and Asia the rates of infertility are much higher, reaching ∼30% in some populations [1,2]. Of note, the real number worldwide is difficult to assess, as there is both heterogeneity in the criteria used to define the disease and in the types of studies employed.”. we have also changed the reference no1 with a more updated one.

INTRODUCTION (note 2)

Please, in brief describe character and function, and current status of the “-omics” approach.

A: We thank you for the comment. We have added some more sentences in the introduction section; it now reads: “Quoting the paper of Altmäe et al (2014), ‘Omics’ refer to the application of high-throughput techniques that simultaneously take under consideration the alterations in the genome, epigenome, transcriptome, proteome or metabolome in a certain biological sample [6]. They are also new fields in biological data, such as exomics, lipidomics, and secretomics. Thus, some of the numerous examples in the literature, are reports centering on creating molecular tools using microarray technology (ERA), on determining the transcriptomic signature of the endometrium during the window of implantation: authors found that 18 of the 25 gene targets were included in the ERA design [8].”.

INTRODUCTION (upgrade) – This papers I consider important to cite

There is a general tendency towards a fully automated performance of clinical analytical protocols and/or management to downstream the overall errors and upstream the quality/quantity of performance under optimal conditions [https://doi.org/10.1016/j.cca.2020.04.015]. Here, the modern IVF Unit is not an exception, therefore several technologies have been proposed and even though they have not yet been fully assessed to be established as reliable approaches in the clinical management of infer-tility, the interest is growing [3].

A: We thank you for the comment. We have added the proposed reference and made the relevant change in the text.

MATERIALS AND METHODS (note)

Please, draw a workflow scheme on the proposed protocol based on coupling the omics with artificial intelligence.

A: We thank you for the comment. A new figure presenting all the components is now added in the manuscript.

Figure 1: Concept diagram of the proposed protocol. Inputs (left side) of the proposed demographics, previous IVF data, and -omics characteristics are used by: a) the statistics processors to extract important features and lead to a reasoning and b) by the various ANNs to predict i) clinical pregnancy, ii) miscarriage and iii) live birth. The ANN system should be able to be used by numerous IVF clinics. IVF clinics issue queries for specific couples and get as response the ANNs outputs at the three prediction levels. Finally the system is supported by a multitier cloud system composed of database, application and web servers, being responsible for: a) store patient data b) run the ANNs and the user logic c) present the results through a web interface.

Reviewer 2 Report

The manuscript is strictly methodological explaining the concept and the design of an artificial intelligence Plattform for fertility consultation IVF. Nevertheless, a graph depicting the designated steps of the design would be useful for the audience of IVF, and also a demonstration of the data input and outcomes would also benefit the reader.

Author Response

We would like to thank the reviewer for taking the time and effort to assess our original submission so meticulously. We have taken into account all of your comments and recommendations and we have modified our paper accordingly. All manuscript changes have been highlighted using the “tracked changes” function provided by Microsoft Word. Detailed replies to the reviewer’s comments are provided below:

The manuscript is strictly methodological explaining the concept and the design of an artificial intelligence Plattform for fertility consultation IVF. Nevertheless, a graph depicting the designated steps of the design would be useful for the audience of IVF, and also a demonstration of the data input and outcomes would also benefit the reader (attached).

A: We thank you for the comment. A new figure presenting all the components is now added in the manuscript.

Figure 1: Concept diagram of the proposed protocol. Inputs (left side) of the proposed demographics, previous IVF data, and -omics characteristics are used by: a) the statistics processors to extract important features and lead to a reasoning and b) by the various ANNs to predict i) clinical pregnancy, ii) miscarriage and iii) live birth. The ANN system should be able to be used by numerous IVF clinics. IVF clinics issue queries for specific couples and get as response the ANNs outputs at the three prediction levels. Finally the system is supported by a multitier cloud system composed of database, application and web servers, being responsible for: a) store patient data b) run the ANNs and the user logic c) present the results through a web interface.

Round 2

Reviewer 1 Report

Authors have accomplished given remarks.